
# Challenges in flood modelling over data scarce regions: how to exploit globally available soil moisture products to estimate antecedent soil wetness conditions in Morocco

El Mahdi El Khalki [1], Yves Tramblay [2,*], Christian Massari [3], Luca Brocca [3], Mohamed El Mehdi Saidi [1]

[1] Geosciences and Environment Laboratory, Cadi Ayyad University, Marrakesh, 40000, Morocco

[2] HydroSciences Montpellier (Univ. Montpellier, CNRS, IRD), Montpellier, 34000, France

[3] Research Institute for Geo-Hydrological Protection, National Research Council, Perugia, 06100, Italy

\* Correspondence to: Yves Tramblay (yves.tramblay@ird.fr)

**Abstract:** The Mediterranean region is characterized by intense rainfall events giving rise to devastating floods. In Maghreb countries such as Morocco, there is a strong need for forecasting systems to reduce the impacts of floods. The development of such a system in the case of ungauged catchments is complicated but remote sensing products could overcome the lack of in-situ measurements. The soil moisture content can strongly modulate the magnitude of flood events and consequently is a crucial parameter to take into account for flood modeling. In this study, different soil moisture products (ESA-CCI, SMOS, SMOS-IC, ASCAT satellite products and ERA5 reanalysis) are compared to in-situ measurements and one continuous soil moisture accounting (SMA) model for basins located in the High-Atlas Mountains, upstream of the city of Marrakech. The results show that the SMOS-IC satellite product and the ERA5 reanalysis are best correlated with observed soil moisture and with the SMA model outputs. The different soil moisture datasets were also compared to estimate the initial soil moisture condition for an event-based hydrological model based on the Soil Conservation Service Curve Number (SCS-CN). The ASCAT, SMOS-IC and ERA5 products performed equally well in validation to simulate floods, outperforming daily in situ soil moisture measurements that may not be representative of the whole catchment soil moisture conditions. The results also indicated that the daily time step may not fully represent the saturation state before a flood event, due to the rapid decay of soil moisture after rainfall in these semi-arid environments. Indeed, at the hourly time step, ERA5 and in-situ measurements were found to better represent the initial soil moisture conditions of the SCS-CN model by comparison with the daily time step. The results of this work could be used to implement efficient flood modelling and forecasting systems in semi-arid regions where soil moisture measurements are lacking.

**Keywords:** Soil moisture, floods, Morocco, ERA5, Rheraya, Issyl, High Atlas


## 1 Introduction

The Mediterranean region is characterized by intense rainfall events generating floods with a very short response time (Gaume et al., 2004; Merheb et al., 2016; Tramblay et al., 2011). The socio-economic consequences of these floods are very important in terms of fatalities or damages to the infrastructures in particular for Southern countries (Vinet et al., 2016). This highlights the need for forecasting systems to reduce the impacts of floods. Unfortunately, the development of such systems is very complicated in the case of ungauged catchments (Creutin and Borga, 2003) such as in North Africa and requires remote sensing products to overcome the lack of in situ measurements. Furthermore, while several studies have been focused on northern Mediterranean catchments for flood modelling, only a few studies are available on southern basins, yet those probably the most vulnerable to floods.

The Moroccan catchments are exposed to intense flash floods, such as the event of August 17, 1995 in the Ourika river where the max discharge reached in 45 minutes a peak discharge of 1030 m3/s causing extensive damages and more than 200 casualties (Saidi et al., 2003). Few studies have been carried out in Morocco to minimize the impact of floods by improving the forecasting systems, either by event-based modeling of floods (El Alaoui El Fels et al., 2017; Boumenni et al., 2017; El Khalki et al., 2018) or by hydro-geomorphological approaches (Bennani et al., 2019) to identify the areas at risk of flooding. The severity of floods in these semi-arid regions is controlled by several factors including precipitation intensity, soil permeability, steep slopes and soil moisture content at the beginning of event (El Khalki et al., 2018; Tramblay et al., 2012). In Mediterranean regions, the soil moisture content varies between events and is known to strongly modulate the magnitude of floods (Brocca et al., 2017; Tuttle and Salvucci, 2014) and particularly to be useful for flood modeling and forecasting systems (Brocca et al., 2011; El Khalki et al., 2018; Koster et al., 2009; Marchandise and Viel, 2010; Tramblay et al., 2012). However, studies in North African basins are lacking to document the rainfall-runoff relationship with soil moisture during floods (Merheb et al., 2016).

In most Mediterranean regions and particularly in North Africa, only a few measurements of soil moisture are available. To represent spatial variability, several measurement at different locations are needed due to the potentially large spatial variability of soil moisture for a wide range of scales (Massari et al., 2014; Schulte et al., 2005; Western and Blöschl, n.d.). However, even the in-situ data may not represent the spatial variability over a very wide area in the case of large basins. On the contrary, satellite soil moisture products provide coverage of the earth's surface by microwave sensors. There are two types of microwave sensors, active and passive, noting: 1) The Advanced Scatterometer (ASCAT) soil moisture product is on board MetOp with good radiometric accuracy and stability. This product provides a spatial resolution of 25 km with a temporal resolution of 1 day since January 2007


(Wagner et al., 2013). 2) The Soil Moisture and Ocean Salinity Mission (SMOS) product, which
begins in January 2010 with a spatial resolution of 50km (Kerr et al., 2012). The improvement of the
robustness of satellite soil moisture products can be achieved by merging passive and active
microwave sensors as initiated and distributed by ESA-CCI (European Space Agency Climate Change
Initiative) (Liu et al., 2011) providing data from 1978 to 2018. However, remote sensing products
might suffer from several problems in complex topography or very dense vegetation and snow cover
(Brocca et al., 2017). For this reason and before any use the data, it is necessary to validate them (Al-
Yaari et al., 2014; Van doninck et al., 2012; Ochsner et al., 2013), either by in-situ measurements, if
they exist, or by using Soil Moisture Accounting models (Javelle et al., 2010; Tramblay et al., 2012) to
simulate soil moisture in the ungauged basins.

In this context, with an increasing number of satellite products becoming available to estimate soil
moisture, clear guidelines and recommendations about the most suitable products to estimate the initial
soil moisture content prior to floods are lacking for the semi-arid basins of North Africa. The purpose
of this study is to compare different satellite soil moisture products with in-situ soil moisture
measurements and the recently developed ERA5 reanalysis to estimate the initial soil moisture before
flood events. The goal is to identify the best products to be used for flood modelling that could
improve forecasting systems. This comparison is performed for two basins representative of medium-
size catchments of North Africa that are the most sensitive to flash flood events. The validation of the
different soil moisture products is made with a Soil Moisture Accounting (SMA) model, to test the
capabilities of the different soil moisture products for the sake of estimating the initial conditions for
an event-based hydrological model for floods. The paper is organized as follow: In section 2, an
overview of the study area and all used data (hydro-meteorological and soil moisture products).
Section 3 explains the methods adopted in this paper. Section 4 presents the results. The conclusion
and perspectives are given in the last section.

**2 Study area and data**

**2.1 Rheraya and Issyl catchments**

The Rheraya research catchment (Jarlan et al., 2015) is located in the Moroccan High Atlas Mountains
(Figure 1) with an altitude ranging from 1027 to 4167m and an area of 225km². The climate in the
basin is semi-arid, strongly influenced by altitude, with a mean annual precipitation of 732mm,
including 30% as snow in altitudes above 2000m (Boudhar et al., 2009). The geology is characterized
by volcanic formations that are considered impermeable in the highest elevation areas, while the
lowest elevation areas are made of granites with clays and marls. In the highest elevation areas very
steep slopes are found with an average of 19% (Chaponnière et al., 2008). The vegetation cover is only



located in the lowest areas with a concentration of cultivated areas found along the river channel.
These natural conditions favor runoff generation. There is very low human disturbance for runoff, with
only some local water uptake in the lower part of the river.

The Issyl basin (Figure 1) is located in the foothills of the Moroccan High Atlas Mountains with an
altitude ranging from 632 to 2300m, an area of 160 km², and a mean annual precipitation of 666mm. It
is an ephemeral river with discharge occurring only after rainfall events. The climate is semi-arid to
arid and the downstream part of the basin reaches the city of Marrakech. The geological formations in
this downstream are alluvial conglomerates that are relatively permeable. The upstream of the basin
consists of clays and calcareous marl. The basin area includes agricultural activities that are irrigated
in the downstream part of the basin. The irrigation comes from *seguias*, earthen-made channels that
traditionally draw their water supply from the river itself, by building small diverting dams on the side
of the river (Pérennès, 1994). The *seguias* channels are usually filled up during floods, and water is
distributed to the neighboring agricultural parcels. The map on the *seguias* in the Issyl basin can be
seen in Figure 1, covering the northern part of the basin. The system is unmonitored and in a context
of high evaporation rates the portion of runoff diverted from the stream is not quantified. Due to the
temporary nature of *seguias*, they can be partially destroyed during large floods and consequently their
hydraulic properties and the amount of water collected can be modified over time.

**2.2 Hydro-meteorological data**

In the Rheraya basin, we used 8 rainfall stations, 5 of them from the data network of the Joint
International Laboratory Télédétection et Ressources en Eau en Méditerranée semi- Aride ''LMI
TREMA'' (Jarlan et al., 2015; Khabba et al., 2013) and the remaining ones from the Tensift Hydraulic
Basin Agency. The data is covering from 2008 to 2016. For the Issyl basin, only 2 rainfall stations are
available from the Tensift Hydraulic Basin Agency, covering the years from 2010 to 2015. In this type
of basin, the spatial variability of rainfall is very important (Chaponnière et al., 2008). The
hydrometric data was provided by radar installed in each basin's outlet. The data is covering only the
year 2014 for Rheraya, since the sensor was installed at the end of 2013, and the years 2010 to 2015
for Issyl. The discharge data is provided with a time step of 10min converted into hourly time step as
for rainfall.

The discharge data is missing in some events that are not selected. For this reason we considered only
the events with complete discharge data. Some of the flood events considered in this study (Table 1)
occurred in winter season, where rainfall can be in the form of snow above 2000m elevation.
According to El khalki et al.(2018) the snow doesn't contribute to runoff during winter season in the
Rheraya basin, where only 17% of basin area is occupied by snow. The runoff coefficients calculated





for each selected events are ranging from 13.1 to 34.1% for Rheraya and from 1.2 to 7.2% for Issyl.
This indicates the important role of initial conditions in both basins, with a much higher infiltration
capacity in the Issyl basin in addition to potential water loss due to irrigation. We used 5 temperature
stations located in the Rheraya basin and one temperature station located in the Issyl basin with an
hourly time step to calculate the average temperature over each basin, ranging from 2008 to 2016. This
data enabled us to calculate potential evapotranspiration (PET) with Oudin formula (Oudin et al.,
2005) requiring temperature only.

**2.3 Soil moisture data**

We used 7 different types of soil moisture data over the Rheraya basin and 6 types in the Issyl basin
due to the absence of measurements in this basin. Covering the same period of rainfall data mentioned
in the 2.3 section, we used:

1.   In-situ measurement with three Thetaprobes at 5cm and 30cm depth in the Rheraya basin,

located at the SMPR7 station (Figure 1).

2.   Simulated soil moisture from a Soil Moisture Accounting model (SMA)

3.   ASCAT satellite soil moisture

4.   SMOS satellite soil moisture

5.   SMOS-IC satellite soil moisture

6.   ESA-CCI satellite soil moisture

7.   ERA5 reanalysis soil moisture


**2.3.1 In-situ measurements**

Soil moisture measurements are available at one location with three Thetaprobes at two different
depths (5cm and 30cm). In this study we used Thetaprobes with 5cm depth, which is comparable with
the depths of  satellite products (Massari et al., 2014).  The site is located in Rheraya basin, with an
altitude of 2030m and a slope of 30% (Figure 1). The data is covering the time period from 2013 to
2016, with 30min time step converted to daily time step.

**2.3.2 Soil moisture accounting model**

The SMA is a continuous Soil Moisture Accounting model that can be used in the absence of soil
moisture data to represent the degree of saturation for flood modeling (Anctil et al., 2004; Tramblay et
al., 2012). In this study, a simplified version of the SMA model is used, adopting the same approach
used by Tramblay et al. (2012) and Javelle et al. (2010). The SMA calculates the level of the soil
reservoir (S/A), ranging between 0 and 1, by calibrating its single parameter, A, which represents the


reservoir capacity. An interpolated daily rainfall dataset created by the Inverse Distance method and
evapotranspiration data computed from daily maximum and minimum temperature with the
Hargreaves-Samani equation (Hargreaves and Samani, 1982) are used as inputs to the SMA model.

**2.3.3 Soil moisture products**

In this study we used three different types of satellite products and a Reanalysis product: an active
product (ASCAT), two variants of a passive product (SMOS and SMOS-IC), a product that combines
the two active and passive products (ESA-CCI) and ERA5 product:

1.  The Advanced SCATterometer (ASCAT) is a Soil Moisture product, onboard Metop-A

and Metop-B and a Metop-C satellite is a C-band (5.255 GHz) scatterometer onboard the

Metop satellite series. It has a spatial sampling of 12.5 km and 1 to 2 observations per day

(Wagner et al., 2013). The SM product was provided within the EUMETSAT project

(http://hsaf.meteoam.it/) denoted as H115.

2.  The Soil Moisture and Ocean Salinity (SMOS) mission  is a radiometer operating at L

band (1.4 GHz), providing Soil Moisture data with ~50km as spatial sampling and 1

observation per 2/3 days (Kerr et al., 2001). Centre Aval de Traitement des Données

SMOS (CATDS, https://www.catds.fr/) provided the version RE04 (level3) for this study.

This version is gridded on the 25km EASEv2 grid.

3.  The Soil Moisture and Ocean Salinity INRA-CESBIO (SMOS-IC) is an algorithm

designed by Insitut National de la Recherche Agronomique (INRA) and Centre d'Etudes

Spatiales de la Biosphère (CESBIO) for a global retrieval of Soil Moisture and L-VOD.

Two parameters of inversion of the L-MED model are used in the SMOS-IC (Wigneron

et al., 2007) with a consideration of the pixel as homogeneous. This version is 105 and

has a spatial sampling of 25km with EASEv2 Grid (Fernandez-Moran et al., 2017).

4.  The ESA-CCI soil moisture product (http://www.esa-soilmoisture-cci.org/) regroups

active and passive microwave sensors to measure soil moisture, giving three type of

products: Active, Passive and Combined (Active + Passive). In this paper, the ESA-CCI

V4.5 – Combined product is used (Dorigo et al., 2017; Gruber et al., 2017, 2019). The

product has been  validated to be useful by 600 ground-based measurement points around

the globe (Dorigo et al., 2015), as well as it was compared with ERA-Interim products

(Albergel et al., 2013). In the field of hydrological modeling, several global studies have

used the ESA-CCI product to initiate the hydrological model (Dorigo et al., 2012, 2015;

Massari et al., 2014) at the scale of Morocco (El Khalki et al., 2018). We extracted for

each basin the pixel that corresponds to it.





5. ERA5 (Copernicus Climate Change Service (C3S), 2017) developed by European Centre for Medium-Range Weather Forecasts (ECMWF), it is the latest version of atmospheric reanalysis available for public since February 2019. The ERA5 replaced ERA-Interim with improvement at different scales, particularly, a higher spatial and temporal resolution, and a better global balance of precipitation and evaporation. The spatial resolution is 31km instead of 79km, hourly resolution is used instead of 6 hours, and the covered period will be extended to 1950 in future. The ERA5 product was applied in some recent studies in hydro-climatic field (Albergel et al., 2018; Hwang et al., 2019; Mahto and Mishra, 2019; Olauson, 2018). We selected the volumetric soil water of the first soil layer. This new product is tested in our study for the first time in Morocco. An alternative dataset, ERA5-Land using an improved land-surface scheme with a spatial resolution of 10km, was also tested, providing the same results as ERA5 since there is a strong correlation between soil moisture simulated by the two products.

## 3 Methods

### 3.1 Evaluation of different soil moisture datasets

In-situ data preparation consists of averaging the 5cm depth probes in order to get a single value to work with and take into account the plot-scale variability of the measurements. This data is considered as a reference for soil moisture data in the Rheraya basin, so that all the other soil moisture products are compared to it. The different soil moisture products are compared to the observed soil moisture over the entire period and also on a seasonal basis.

The SMA model is used to represent the soil moisture aggregated at the catchment scale. The rationale behind the use of such model here is that continuous rainfall and temperature series are often available in monitored catchments, unlike soil moisture, and a calibrated SMA model can sometimes palliate the lack of soil moisture measurements (Tramblay et al., 2012). For the SMA model, the A parameter, representing the soil water holding capacity, is calibrated to obtain the best correlation between observed and simulated soil moisture (S/A). The calibration with observed data can only be performed in the Rheraya basin where soil moisture is measured. In addition to this calibration, other values of A, ranging between 1 and 1000, are tested in the SMA model to maximize the correlations with the different soil moisture products. The choice of this approach is to check if there are any possible uncertainties that can be related to the in-situ soil moisture measurements, located on a steep slope plot that may not fully represent the average soil moisture conditions over the whole basin. In the case of the Issyl basin, since there is no observed soil moisture data, the model is run for a range of different





values of the A parameter. The best value of the A parameter is selected as the one yielding the best
correlations with the different satellite products.

The values from ASCAT and SMA are given in percentage (values are ranging between 0 and 1) while
SMOS, SMOS-IC, ERA5, ESA-CCI and observations are in $m^3\ m^{-3}$. To allow a comparison for all soil
moisture datasets a rescaling procedure is needed. Before applying the rescaling procedure, according
to Albergel et al. (2010), a 95% confidence interval is chosen to define the higher and lower values to
exclude any abnormal outliers using equation 1 and 2. The resulted data is then rescaled to their own
maximum and minimum values considering the whole period using the equation 3. The issue in the
validation of satellite soil moisture products and reanalysis product with in-situ measurements is the
spatial resolution (Jackson et al., 2010). Several studies mentioned that, in the case of the temporal
stability  introduced by Vachaud et al. (1985), one in-situ measurement point can represent the soil
moisture condition of a larger area (Brocca et al., 2009b, 2010; Loew and Mauser, 2008; Loew and
Schlenz, 2011; Martínez-Fernández and Ceballos, 2005; Miralles et al., 2010; Wagner et al., 2008).
According to (Massari et al., 2015), the coarse satellite observations can be beneficial for small basins,
in the case if the in-situ observation falls in the satellite product pixel. This means that the in-situ
measurements can represent a good benchmark (Liu et al., 2011). In this study we considered the in-
situ measurement as a benchmark to validate different soil moisture products.

$$Up_{SM} = \mu_{SM} + 1.96\sigma_{SM}, \qquad (1)$$
$$Low_{SM} = \mu_{SM} - 1.96\sigma_{SM}, \qquad (2)$$

Where $Up_{SM}$ and $Low_{SM}$ are the limits of the confidence interval (the upper and the lower 95%)

$$SM = \frac{SM - Low_{SM}}{Low_{SM} - Up_{SM}}, \qquad (3)$$


**3.2 Extended collocation analysis:**

An alternative technique to validate soil moisture products when ground truth is missing is the use of
Triple Collocation (TC) analysis (Gruber et al. 2016b). TC analysis requires the availability of three
datasets with mutually independent errors and linear additive error model between the measurement
systems and the unknown truth:

$$X = \alpha + \beta S + \varepsilon, \qquad (4)$$


where X is the soil moisture estimate, S is the true soil moisture, α and β are additive and
multiplicative biases, respectively. Eventually, ε is the zero-mean random error.






To build such a triplet, satellite and ground-based datasets can be combined with modeled soil
moisture fields from reanalysis (e.g., ERA5). The reanalysis datasets ingest a number of satellite,
atmospheric and ground observations which can potentially undermine their independence with
respect to other members of the triplets. This creates doubts about the satisfaction of the null cross-
correlation assumptions required to apply TC (Stoffelen, 1998). In a preliminary analysis (not shown),
we used TC to characterize the error variance of the different soil moisture datasets by using different
triplet combinations of the products. However, we observed substantial differences among the selected
triplets likely due to error co-dependence. Based on that, we assumed the existence of non-null error
cross correlation for the selected triplets (e.g. ERA5, SMOS and ASCAT).

When more than three products are available (i.e., N), the error can be estimated using an Extended
Collocation (EC) approach (Gruber et al. 2016). The same assumptions for TC also apply for EC, but
the number (N >3) datasets constitutes an over-constrained system, allowing the designation of N-3
non-zero error covariance terms which can be estimated with a least-squares solution (Pierdicca et al.
2015). Therefore, the zero TC assumption can be relaxed to allow non-zero correlation among N-3
data product pairs. For N = 4, the X, Y, Z, W measurement systems and assuming that non-zero EC
exists only between X and Y, the least-squares solution for the QC problem is given by:

$$
M = \begin{bmatrix} \sigma_X^2 \\ \sigma_Y^2 \\ \sigma_Z^2 \\ \sigma_W^2 \\ \sigma_{XY} \\ \sigma_{XZ}\sigma_{XW}/\sigma_{ZW} \\ \sigma_{YZ}\sigma_{YW}/\sigma_{ZW} \\ \sigma_{XZ}\sigma_{ZW}/\sigma_{XW} \\ \sigma_{YZ}\sigma_{ZW}/\sigma_{YW} \\ \sigma_{XW}\sigma_{ZW}/\sigma_{XZ} \\ \sigma_{YW}\sigma_{ZW}/\sigma_{YZ} \\ \sigma_{XZ}\sigma_{YW}/\sigma_{ZW} \\ \sigma_{XW}\sigma_{YZ}/\sigma_{ZW} \end{bmatrix} \quad A = \begin{bmatrix} 1000010000 \\ 0100001000 \\ 0010000100 \\ 0001000010 \\ 0000100001 \\ 1000000000 \\ 0100000000 \\ 0010000000 \\ 0010000000 \\ 0001000000 \\ 0001000000 \\ 0000100000 \\ 0000100000 \end{bmatrix} \quad S = \begin{bmatrix} \beta_X^2\sigma_T^2 \\ \beta_Y^2\sigma_T^2 \\ \beta_Z^2\sigma_T^2 \\ \beta_W^2\sigma_T^2 \\ \beta_X\beta_Y\sigma_T^2 \\ \sigma_{\varepsilon_X}^2 \\ \sigma_{\varepsilon_Y}^2 \\ \sigma_{\varepsilon_Z}^2 \\ \sigma_{\varepsilon_W}^2 \\ \sigma_{\varepsilon_X\varepsilon_Y} \end{bmatrix},
$$

(5)


where $\sigma\_T^2$ is the true soil moisture variance, $\sigma\_\varepsilon^2$ is the variance of the random error, and $\sigma\_{(\varepsilon\_X}$
$\varepsilon\_Y)$ is the error covariance between X and Y.

And the least squares solution for the parameters in S is given as:

$$\hat{S} = (A^T A)^{-1} A^T M,$$

(6)



Which provide the error variance of each dataset as long as the error covariance terms. More details on
the method and its mathematical derivation can be found in Gruber et al. (2016).The error variance
provided by EC can also be expressed in normalised form as Signal-to-Noise Ratio (SNR). This
overcomes the dependency on the chosen scaling reference and allows to compare the error variances
between the data sets. SNR is usually given in decibel, which can be easily interpreted: a value of zero
means that the signal variance is equal to the noise variance, and every 3dB increase(decrease) implies
a doubling (halving) of the signal variance compared to the noise variance. The SNR (expressed in
dB) can be computed using the following formulation:

$$SNR[db] = 10 \log \frac{\beta_i^2 \sigma_\theta^2}{MSE_i},$$ (7)

with i, j in [X, Y, Z] and i ≠ j.

In some special cases, the $MSE_i$ can become negative and the SNR cannot be expressed in dB
(logarithm of a negative number is undefined). The reason is that the relation of the covariances
between the data sets become larger than the actual signal variance (e.g. #XY #XZ/#Y Z > #2X),
which can be related numerical problems, wrong estimation of the covariances or a violation of the
underlying assumptions of the error model in general.In our study we used two different
configurations of the EC techniques. In particular, for the Issyl basin no in situ observations are
available so we used quadruple collocation analysis with quadruplets constructed with ASCAT,
SMOS, ERA5 and SMA and ASCAT, SMOS-IC, ERA5 and SMA. The choice of these quadruplets
was based on the assumption of non-zero correlation between SMOS products and ERA5 so in the
process we also estimated  σ_(SMOS-ERA) (not shown). Similarly, for Rheraya we applied the
methods by using five different datasets and assuming SMOS and ERA products and SMA and in situ
observations characterized by non-null error cross-correlations. For both basins we used either SMOS
or SMOS-IC in the configurations.

**3.3 Event-based hydrological model for floods**

In this study, we used the Soil Conservation Service Curve Number (SCS-CN) model for each basin,
implemented in the hydrologic Engineering System - Hydrologic Modeling System ''HEC-HMS''
software (US Army Corps of Engineers, 2015). This model is known by its widespread popularity  and
to the simplicity of the application method (Miliani et al., 2011). SCS-CN is often used in the semi-
arid context (Brocca et al., 2009a; El Khalki et al., 2018; Tramblay et al., 2010; Zema et al., 2017).
Our methodology is based on the use of SCS-CN model as a production function to compute net
rainfall, by manually calibrating the Curve Number parameter (CN), the value of CN is non-



dimensional ranging from 0 (dry) to 100 (wet). The potential maximum retention, S, is related to CN
as follows:

$$S = \frac{25400}{CN} - 254 \,, \tag{8}$$


The transformation of precipitation excess into runoff is provided by Clark Unit hydrograph model
(transfer function). The calibration procedure is based on calibrating the Clark Unit hydrograph model
parameters; Storage Coefficient (Sc) and Time of Concentration (Tc). The two functions (production
and transfer) are calibrated separately to avoid the parameter dependence.

The validation procedure is based on two steps; first, testing the relationship between soil moisture
data (In-situ, SMA, ERA5, ASCAT, SMOS, SMOS-IC and ESA-CCI), at two different timescales
(daily and hourly) and the S parameter of the event-based model of all the flood events.  The hourly
time step concerns only the in-situ data and ERA5 by choosing the soil moisture state 1 hour before
the starting time of rainfall for each event. Only the ERA5 product can be used in the Issyl basin at the
hourly time step due to the absence of observed data. Then, the soil moisture products that are well
correlated with S parameter are used to validate the model by calculating the S parameter from the
linear equation obtained between soil moisture and S, using the leave-one-out resampling procedure;
each event is successively removed and a new relationship between the remaining event is re-
computed. The estimated S parameter for a given event is then used in the SCS-CN model in
validation. For the Clark Unit Hydrograph model, the average of the Sc and the Tc parameters are used
in validation.

The correlation coefficient of Pearson equation (9) and the Root Mean Square Deviation (RMSD)
equation (10) are used to compare in-situ measurements and humidity modeled by SMA model and
the different soil moisture products. For the evaluation of the flows simulated by the flood event
model, we compared the simulated discharge with those observed using the efficiency coefficient of
Nash-Sutcliffe (Ns) (Nash and Sutcliffe, 1970) equation (11) as well as through the bias on peak flow
and on volume equation(12).

$$r = \frac{N \sum SM_{sat} SM_{In-situ} - (\sum SM_{sat})(\sum SM_{In-situ})}{\sqrt{[N \sum SM_{sat}^2 - (\sum SM_{sat})^2][N \sum SM_{In-situ}^2 - (\sum SM_{In-situ})^2]}}, \tag{9}$$

$$RMSD = \sqrt{\frac{\sum_{i=1}^{n}(SM_{In-situ} - SM_{sat})^2}{N}}, \tag{10}$$

$$Ns = 1 - \frac{\sum_{i=1}^{n}(Q_{obs,i} - Q_{sim,i})^2}{\sum_{i=1}^{n}(Q_{obs,i} - \overline{Q_{obs}})^2}, \tag{11}$$





$$\text{BIAS}_Q = \frac{(Q_{sim} - Q_{obs})}{Q_{obs}}, \tag{12}$$


Where $Q_{sim}$ is the simulated discharge, $Q_{obs}$ is the observed discharge, $SM_{In-situ}$ is the in-situ
measurements of soil moisture, $SM_{sat}$ is the soil moisture from satellite or reanalysis and N is the
number of values. The Ns ranges between -∞ and 1, the 1 value of Ns indicates that the simulated
discharge perfectly match the observed hydrograph

**4 Results and discussions**

**4.1 Relationship between satellite soil moisture data and in-situ measurements**

The comparison between measured soil moisture at 5cm depth and the different products of soil
moisture show that the SMOS-IC and ERA5 provide the best correlations, with r=0.76 and r=0.67
respectively, but it should be noted that all the correlations with the different products are also
significant. Figure 2 shows that SMOS-IC and ERA5 reproduce dry periods well, whereas ERA5
reproduces well wet periods. This result is in accordance with the results of Massari et al. (2014) who
found that ERA-Land is well correlated with In-situ data. ASCAT product shows a correlation of
r=0.43 which is less than the correlation given in Albergel et al. (2010) who found r values ranging
from between 0.59 and 0.64, the lower correlation may be caused by the orography and the coarse
resolution. In fact, this results shows that the use of a combined product as ESA-CCI give an obvious
advances in term of r values than one single satellite soil moisture product (Ma et al., 2019; Zeng et
al., 2015). It should be noted that the soil moisture products have a different percentage of missing
data for ASCAT (0%), SMOS (18.7%), SMOS-IC (6.82%), ESA-CCI (46%) and observed soil
moisture (12%). The ESA-CCI showed an important percentage of missing values comparing to
ASCAT that is integrated in the ESA-CCI product. This due to the filter used in the ESA-CCI product
to ensure the data quality, more description can be found in (Dorigo et al., 2017).

**4.2 Relationship between the SMA model outputs and soil moisture products**

The best correlation between observed soil moisture and the soil moisture level (S/A) modeled by the
SMA model is obtained for A=8mm with r=0.86. But it shows higher RMSD than observations
(RMSD =0.23) which is due to the overestimation of the wet periods (Figure 3). This can be related to
the averaging of rainfall data in the SMA model over the basin which could be higher than rainfall in
the soil moisture measurement site. It should be noted that the value of the A parameter is very small
by comparing to previous studies (Javelle et al., 2010; Tramblay et al., 2012), indicating a much lower
soil storage capacity.






We correlated the SMA model output (for A=8mm) with the Satellite Products of Soil Moisture, and
the best correlations are found for SMOS-IC and ERA-5, with r=0.74 and r=0.63 respectively (Figure
4). Other values of A that maximize the correlations with the different soil moisture products have also
been tested. Optimal values of A are ranging from 1 mm with ASCAT (with r= 0.4), 8 mm for SMOS
(r=0.56), SMOS-IC (r=0.75) and ESA-CCI (r=0.55) up to 16mm for ERA5 (r=0.68). Comparing the
Figure 2 and Figure 4 we notice that the soil moisture products better reproduce in-situ measurements
than modelled soil moisture with the SMA model, expect for ESA-CCI and SMOS. This improvement
is directly related to the SMA model performance, which overestimates soil moisture, and should be
compared to Figure 2 where ESA-CCI and SMOS products also overestimate in-situ measurements.

For the Issyl basin, the percentage of missing values is a bit lower than in the Rheraya and also
different between the satellite products: ASCAT (0%), SMOS (17.19%), SMOS-IC (9.1%) and ESA-
CCI (2.2%). As mentioned above, no observed soil moisture data is available in the Issyl basin to
calibrate the A parameter of the SMA model. Therefore, different values of A are tested to correlate
the SMA outputs with the different soil moisture datasets. Over all datasets, the value of A best
correlated to the majority of soil moisture products is 30mm. The best correlation is given by
A=30mm with r=0.78, 0.82 and 0.79 for ASCAT, SMOS-IC and ESA-CCI respectively. As for SMOS
and ERA5, the best correlation is given for A=40mm with r=0.7 and A=60mm with r=0.8,
respectively. In order to choose a single value of A that represents the basin, we have considered
A=30mm, the optimal value yielding the best correlations with the different soil moisture products.
Figure 5 shows that the best correlation between satellite products and S/A is obtained with SMOS-IC
(r=0.82) and ESA-CCI (r=0.79). As observed over the Rheraya basin, the SMOS-IC and ERA5
products showed a good reproduction for dry periods with a better reproduction of wet periods with
ERA5, these results are similar to those of Ma et al. (2019) who found that SMOS-IC performs well in
arid zones with a median r value of 0.6. Overall, the higher value for the A parameter found for this
basin is coherent with the fact that this basin is located in a plain area with a much higher soil moisture
storage capacity than in the mountainous Rheraya basin.

**4.3. Comparison of soil moisture datasets by seasons**

Seasonal evaluation of satellite soil moisture and reanalysis data shows for the Rheraya basin that
during the summer season there are low correlations (average r=0.34) for all the products which is
possibly due to very low precipitation amounts mostly as localized convective precipitation (Albergel
et al., 2010). On the contrary, better performance are obtained with the SMA model (r=0.59) that
considers catchment-scale precipitations. Better correlations are obtained in fall with an average of
r=0.61 and 0.58 for the in-situ data and SMA respectively (Table 2). In the winter we found a poor


correlation using SMOS and ESA-CCI that can be related to the important percentage of missing
values. For the Issyl watershed, the satellite products show good correlations with the SMA model
outputs (on average r=0.76) except for the SMOS product especially in winter. We also notice a trend
of improving correlations by moving from winter to autumn with a similarity between spring and
autumn, which is not the case in the Rheraya basin, probably because of different precipitation
patterns. The ERA5 overall product shows good correlations for most seasons.

**4.4 Extended collocation analysis**

Table 3 shows the results obtained for the two basins and two configurations. For Issyl, it can be seen
that SMOS-IC is the best performing product with SNR much larger 3DB, followed by ASCAT and
SMA. Conversely ERA5 and SMOS are suboptimal having noise variance similar to the signal
variance. For Rheraya SMOS-IC is the only product providing SNR>3DB followed by SMOS and
ERA5 which are however are still suboptimal. Poor results are found for both SMA, in situ and
ASCAT in this catchment. Overall, the results of this complementary analysis confirm the findings of
previous sections.

**4.5 Calibration of the event-based hydrological model**

Calibration results (Table 4) on the individual flood events of Table 1 show that the difference
between the values of the potential maximum soil moisture retention (S) of each basin is very
important with larger values for the Issyl basin where the soil depth is prominent. We noticed that the
temporal variability of soil moisture can be important between two successive events like the events of
02/04/2012 and 05/04/2012 for the Issyl basin. The SCS-CN model reproduces well the floods of the
Rheraya basin with average Ns of 0.67 and bias on runoff peak ($BIAS_Q$) of 4% (Table 3). As shown on
Figure 6, the SCS-CN model in calibration is able to reproduce the shape of the different flood events
even for the most complex ones (21/04/2014 and 22/11/2014). Similarly, for the Issyl basin the SCS-
CN model gives good results with average Ns of 0.66 and an average bias on runoff peak of 6.93%.
Figure 7 shows the simulated hydrographs which are in good agreement with the observations. The
lower Ns coefficients obtained for the 23/01/2014 event in the Rheraya and for the 03/04/2011 and
28/09/2012 events in the Issyl basin are caused by a slight shift in the hydrograph probably due to a
time lag in instantaneous precipitation measurements. For the Clark Unit Hydrograph model, the
averages of calibrated Tc and Sc parameters are considered for validation (Sc = 1.42 and 2.54 hours
and Tc = 2.85 and 3.64 hours for Rheraya and Issyl respectively).

The S parameters of the hydrological models, for the two basins, are then compared to the soil
moisture products. For the Rheraya basin, there are significant correlations of the S parameter with in-
situ soil moisture data, ERA5 and SMOS-IC (Table 5). The correlations using observed soil moisture,
ESA-CCI and SMOS data can be computed with only 8 and 6 events respectively, due to the presence
of missing values. The time step of the soil moisture data in the Rheraya basin seems to play a key role
in the representation of soil moisture conditions. Indeed, the daily time step shows a weakness to
effectively represent the antecedent soil moisture conditions in the SCS model, which indicates the
rapid change of soil moisture content in such a semi-arid mountainous basin. For the Issyl basin, ESA-
CCI is the only satellite product that is significantly correlated to the S parameter at the daily time
step. The ERA5 product is also significantly correlated with the S parameter but at the hourly time
step. The daily output of the SMA model is also able to estimate the initial condition of the model for
the Issyl basin, with a correlation of -0.69 with S. Interestingly, the SMA model does not provide a
good performance in the Rheraya basin. It can be due to the fact that in such a mountainous basin,
there is a strong spatial variability of rainfall and it is difficult to obtain reliable precipitation estimates
for continuous simulations (Chapponiere et al., 2005).

**4.6 Validation of the event-based hydrological model**

The validation of the event-based hydrological model is performed on the events of Rheraya and Issyl
using only the soil moisture datasets that show relatively good correlations with the initial condition
(S) of the model from Table 6. These products include SMOS-IC, ERA5 and observed soil moisture
for the Rheraya, and ESA-CCI, ERA5, SMOS and SMA for Issyl. The validation of the event-based
model is performed with S calculated from the linear equation obtained from the correlation analysis
between the different soil moisture products and the calibrated parameter S. The validation results
show that for the Rheraya basin the events are well validated using both daily (Figure 6) and hourly
(Figure 7) time step of soil moisture products. The best validation result at the daily time step is
obtained with SMOS-IC with an average Ns of 0.58 for all events (median Ns =0.63). This result
should be compared with the results found in the previous sections where SMOS-IC showed the best
correlations with observed soil moisture. ASCAT and ERA5 show similar results in term of average
Ns (~0.45). On the contrary, the daily observed soil moisture shows a lower performance with an
average Ns of 0.25 (median Ns =0.49). The hourly time step enhanced the performance of the model,
with an average Ns using the ERA5 product of 0.64 (median Ns = 0.73) and also a better performance
with the hourly in-situ data with mean Ns = 0.54 (median Ns = 0.61). These results show that the
hourly time step better represents the saturation content before the flood events in this bassin. For the
Issyl, the validation results are quite different (Figure 8). For only 5 events (the 03/04/2011,
02/05/2011, 19/05/2011, 05/04/2012 and 25/03/2015) the event-based model can be validated using
the ERA5 hourly data with an average Ns coefficient of 0.46, while for all other events and with
different soil moisture products the Ns coefficients are negative and the hydrographs not adequately
reproduced. These validation results should be put in perspective with the fact that the Issyl basin has a





land use characterized by agricultural activities with possible large water uptake in the diver channel
during floods for irrigation. Some simple methods to compensate for the water losses due to irrigation,
such as the application of a varying percentage of runoff added to the observed discharge to
compensate the part of water lost for irrigation, have been tested but with no improvement of the
results. This is probably because the quantity taken for irrigation is not constant from one event to
another depending on the farmer needs, as shown by field surveys, and this amount may also depend
on discharge thresholds.

**5 Conclusions**

This study performed an evaluation of different soil moisture products (ASCAT, ESA-CCI, SMOS,
SMOS-IC and ERA5) using in-situ measurements and a Soil Moisture Accounting model (SMA) over
two basins located in the Moroccan High Atlas in order to estimate the initial soil moisture conditions
before flood events. There is a knowledge gap on the evaluation of soil moisture products in North
Africa (Jiang and Wang, 2019) that the present study aimed to fill. The results indicated that the
SMOS-IC product is well correlated with both the in-situ soil moisture measurements and simulated
soil moisture from the SMA model over the two basins. Beside satellite products, the new ERA5
reanalysis reproduced also well the in-situ measurements over the mountainous basin, which indicates
the robustness of this product to estimate soil moisture in these semi-arid environments. The seasonal
analysis showed increasing correlations coefficients, from winter to autumn, for all the soil moisture
products when compared to observations, which encourages the use of these remote sensing products
for flood forecasting because the majority of events occur in autumn and early winter in these regions
(El Khalki et al., 2018). The extended collocation analysis show coherent results with the correlation
results with the SMOS-IC providing the best results for the Issyl and Rheraya basins. One of the main
finding of the present study is that different products, in particular SMOS-IC, ASCAT and ERA5, are
efficient to estimate the initial soil moisture conditions in an event-based hydrological model, that
could improve the forecasting capability in data-scare environments.

This study also showed that the hourly temporal resolution for soil moisture may provide a better
estimate of the initial soil moisture conditions for both basins. Indeed, the use of hourly in-situ soil
moisture measurements and ERA5 provided better performance to estimate the initial condition of the
hydrological model. These results indicate that the temporal variability of soil moisture in these semi-
arid basins under high evapotranspiration rates can be very important causing a quick decay of soil
moisture following a rainfall event. For this type of basin or others under even more arid conditions,
the use of soil moisture products with an hourly temporal resolution could be required to estimate with
accuracy the soil moisture content prior to flood events. This constitute a research challenge to
monitor soil moisture at the sub-daily timescale without ground measurements, since most remote



sensing products at present are not available at the hourly time step. As shown by this study,
atmospheric reanalysis coupled with a land surface model, such as ERA5, could provide a valuable
alternative, in particular since the resolution of these products is constantly improving along with a
more realistic representation of water balance.

For the catchment that is the most influenced by agricultural activities, the Issyl basin located nearby
Marrakech, the water uptake for irrigation made difficult the validation of the hydrological model. The
model overestimates runoff for some flood events, since the water uptake during floods from the river
channel by small artisanal structures is not monitored and thus cannot be represented in the
hydrological model. This example show the difficulty in the implementation of a flood forecasting
system in such basin without a good knowledge on the human influences on river discharge. This
situation is not a particular case but deemed common in semi-arid areas where rivers with a high risk
of flooding are also a substantial water resource for agriculture. Therefore, as shown by our results, a
hydrological model that is not accounting for water use and irrigation may not be efficient at
reproducing flood events in an operational context. The resolution of this issue would requires the
development of an irrigation monitoring system, that would need intensive field surveys and mapping
but also the agreement of the local farmers that benefit from this system.

This study is a first step towards the development of operational flood forecasting systems in semi-arid
North Africa basins highly impacted by floods. Indeed, the evaluation of the most suitable satellite or
reanalysis products to estimate soil moisture for the monitoring of the basin saturation conditions
before floods is a necessary first step prior to implement flood warning systems based on rainfall and
soil moisture thresholds or coupled hydrometeorological modelling (Javelle et al., 2010; Norbiato et al.,
2008). One important aspect that should be addressed in further research aiming at developing a flood
forecasting system is the selection of soil moisture data based on the latency of these products. For
instance the ERA5 reanalysis is available within 5-days latency when ASCAT or SMOS satellite
products could be available with 3-hours latency. Prior to these developments, this type of evaluation
should be generalized in Morocco and other sites in North Africa where soil moisture measurements
are available, for the development of reliable flood forecasting systems using the outputs of
meteorological models in combination with the soil moisture state.

**Author Contributions:** E.E.; performed the analysis and wrote the paper, Y.T.; designed the analysis
and wrote the paper, C.M. and L.B.; designed the analysis and contributed to the paper, C.M.
performed the TC analysis, and M.S.; contributed to the paper

**Acknowledgments:** This research has been conducted in TREMA International Joint Laboratory
(https://www.lmi-trema.ma/) funded by the University Cadi Ayyad of Marrakech and the French IRD.





This work is a contribution to the HYdrological cycle in The Mediterranean EXperiment (HyMeX)
program, through INSU-MISTRALS support. The financial support provided by the ERASMUS+
mobility and the Centre National de la Recherche Scientifique et Technique (CNRST) are gratefully
acknowledged. Thanks are due to the hydrological basin agency Tensift (ABHT) and to the LMI
TREMA for providing the data. The Authors would like to thank Professor Khalid Chaouch for his
English revision.

**Conflicts of Interest:** The authors declare no conflict of interest.

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



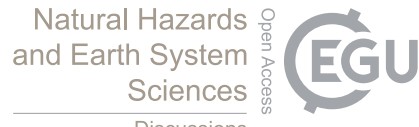



















**TABLES**


**Table 1: Characteristics of the selected flood events.**

| | Rheraya | | | |
|---|---|---|---|---|
| | Max Discharge [m³/s] | Volume [10³ m³] | Precipitation Volume [10³ m³] | Runoff Coefficient [%] |
| 23/01/2014 | 17.1 | 459.2 | 2749.5 | 16.7 |
| 29/01/2014 | 39.7 | 602.8 | 2632.5 | 22.9 |
| 10/02/2014 | 19.2 | 543.2 | 2904.7 | 18.7 |
| 11/03/2014 | 19 | 557 | 1633.5 | 34.1 |
| 21/04/2014 | 38.2 | 1070 | 5431.5 | 19.7 |
| 21/09/2014 | 24.4 | 440.6 | 3363.8 | 13.1 |
| 05/11/2014 | 46.5 | 1027 | 5737.5 | 17.9 |
| 09/11/2014 | 42.2 | 869.3 | 4575.2 | 19 |
| 22/11/2014 | 99.5 | 3868.9 | 17586 | 22 |
| 28/11/2014 | 76.4 | 3797.2 | 11940.8 | 31.8 |
| | Issyl | | | |
| 25/03/2011 | 63.8 | 385.28 | 27520 | 1.4 |



| 03/04/2011 | 16.6 | 550.656 | 30592 | 1.8 |
|---|---|---|---|---|
| 29/04/2011 | 19.7 | 246.4 | 11200 | 2.2 |
| 02/05/2011 | 17.1 | 303.36 | 10112 | 3.0 |
| 16/05/2011 | 45.8 | 361.12 | 9760 | 3.7 |
| 19/05/2011 | 27.6 | 315.392 | 7168 | 4.4 |
| 06/06/2011 | 18.3 | 212.352 | 5056 | 4.2 |
| 02/04/2012 | 16.8 | 216.576 | 18048 | 1.2 |
| 05/04/2012 | 20 | 543.744 | 7552 | 7.2 |
| 28/09/2012 | 22.7 | 126.72 | 7040 | 1.8 |
| 05/04/2013 | 15.4 | 365.376 | 16608 | 2.2 |
| 28/11/2014 | 37.2 | 489.6 | 28800 | 1.7 |
| 25/03/2015 | 16.2 | 767.424 | 18272 | 4.2 |






**Table 2: Results of correlation analysis between soil moisture data and in-situ measurements**
**and SMA model (significant correlations are represented in bold)**

| | | Winter | Spring | Summer | Fall |
|---|---|---|---|---|---|
| | | **Rheraya** | | | |
| In-situ | SMA A=8mm | **0.82** | **0.83** | **0.67** | **0.75** |
| ASCAT | In-situ | **0.47** | -0.03 | 0.18 | **0.70** |
| | SMA A=8mm | **0.32** | 0.09 | **0.54** | **0.65** |
| SMOS | In-situ | 0.01 | **0.68** | **0.61** | 0.16 |
| | SMA A=8mm | -0.09 | **0.75** | **0.58** | **0.54** |
| SMOS-IC | In-situ | **0.80** | **0.68** | **0.45** | **0.85** |
| | SMA A=8mm | **0.80** | **0.72** | **0.62** | **0.57** |
| ESACCI | In-situ | 0.12 | 0.28 | **0.41** | **0.60** |
| | SMA A=8mm | 0.15 | **0.30** | **0.67** | **0.51** |
| ERA5 | In-situ | **0.74** | **0.73** | 0.04 | **0.73** |
| | SMA A=8mm | **0.86** | **0.76** | **0.54** | **0.65** |
| Mean | In-situ | 0.43 | 0.47 | 0.34 | 0.61 |
| | SMA A=8mm | 0.41 | 0.52 | 0.59 | 0.58 |
| | | **Issyl** | | | |
| ASCAT | SMA A=30mm | **0.77** | **0.86** | **0.70** | **0.90** |
| SMOS | | **0.39** | **0.76** | **0.47** | **0.74** |




| | | | | | |
|---|---|---|---|---|---|
| SMOS-IC | | **0.85** | **0.81** | **0.56** | **0.93** |
| ESACCI | | **0.70** | **0.89** | **0.77** | **0.89** |
| ERA5 | | **0.88** | **0.82** | **0.70** | **0.88** |
| Mean | SMA A=30mm | 0.72 | 0.83 | 0.64 | 0.87 |



**Table 3: Signal to noise ratio for Rheraya and Issyl basins. The SNT = 0 : Error variance, SNR > 3 Signal**
**variance double the noice variance (very good) and SNR < 3 Signal variance half noice variance (not**
**good).**

| | ASCAT | SMOS | SMOS-IC | ERA5 | SMA |
|---|---|---|---|---|---|
| **Rheraya** | -5.55 | | 7.54 | | -1.99 |
| | -6.16 | 4.31 | | 1.16 | -1.10 |
| **Issyl** | 4.23 | 1.90 | | 2.33 | 5.03 |
| | 4.28 | | 8.12 | 2.33 | 4.99 |







**Table 4: Calibration results of SCS-CN model, S is the potential maximum soil moisture retention,**
**$BIAS_Q$ is the difference between the observed and calibrated peak discharge of the event, $BIAS_V$ is**
**the difference between the observed and calibrated volume of the event.**

| Rheraya | | | | | Issyl | | | | |
|---|---|---|---|---|---|---|---|---|---|
| Events | S[mm] | Ns | $BIAS_Q$ [%] | $BIAS_V$ [%] | Events | S[mm] | Ns | $BIAS_Q$ [%] | $BIAS_V$ [%] |
| 23/01/2014 | 19.1 | -0.58 | 1.18 | -5.76 | 25/03/2011 | 679.8 | 0,83 | 29,94 | -13,5 |
| 29/01/2014 | 24.5 | 0.87 | 6.43 | 29.14 | 03/04/2011 | 730.5 | 0,02 | -12,05 | 27,93 |
| 10/02/2014 | 34.6 | 0.71 | -4 | 2.85 | 29/04/2011 | 218.1 | 0,83 | 0 | 10,36 |
| 11/03/2014 | 9.5 | 0.61 | -17.39 | 2.57 | 02/05/2011 | 113 | 0,91 | -0,58 | 44,39 |
| 21/04/2014 | 55.8 | 0.73 | 6.41 | 2.3 | 16/05/2011 | 176.5 | 0,61 | 17,69 | -26,31 |
| 21/09/2014 | 34.6 | 0.77 | 27.08 | -6.87 | 19/05/2011 | 136.7 | 0,87 | 1,09 | 9,64 |
| 05/11/2014 | 39.6 | 0.97 | 15.38 | 0.88 | 06/06/2011 | 108.8 | 0,75 | 0 | -5,38 |
| 09/11/2014 | 40.7 | 0.83 | 6.3 | -0.32 | 02/04/2012 | 440.3 | 0,56 | 0 | 15,26 |
| 22/11/2014 | 43.1 | 0.78 | -5.06 | 2.38 | 05/04/2012 | 125.1 | 0,56 | 13,5 | -1,91 |
| 28/11/2014 | 71.6 | 0.97 | 3.66 | -6.22 | 28/09/2012 | 159.7 | 0,11 | 32,16 | 23,41 |
| | | | | | 05/04/2013 | 388.2 | 0,9 | 6,49 | -4,16 |





| | | | | | | | | |
|---|---|---|---|---|---|---|---|---|
| | | | | 28/11/2014 | 254 | 0,74 | 1,88 | 0,71 |
| | | | | 25/03/2015 | 356.6 | 0,89 | 0 | 12,32 |
| Mean | 0.67 | 4 | 2.09 | Mean | | 0,66 | 6,93 | 7,14 |
| Median | 0.77 | 4.98 | 1.59 | Median | | 0,75 | 1,09 | 9,64 |














**Table 5: Correlation between soil moisture products and the S parameter of the SCS-CN**
**hydrological model**

| | Rheraya | | Issyl | |
|---|---|---|---|---|
| | **S** | **Number of events** | **S** | **Number of events** |
| In-situ [Daily] | -0.71 | 8 | - | - |
| In-situ [Hourly] | -0.83 | 8 | - | - |
| SMA A=8mm | -0.32 | 10 | - | - |
| SMA A=30mm | 0.02 | 10 | -0.69 | 13 |
| ASCAT | -0.55 | 10 | -0,29 | 13 |
| ESA-CCI | -0,29 | 8 | -0.66 | 11 |
| SMOS | 0.12 | 6 | -0,59 | 6 |
| SMOS-IC | -0.81 | 10 | -0.34 | 13 |
| ERA5 [Daily] | -0.46 | 10 | -0.37 | 13 |
| ERA5 [Hourly] | -0.80 | 10 | -0.63 | 13 |






**Table 6: Performance of the SCS-CN model in term of Nash Coefficient for the Rheraya and Issyl events,**
**using the daily or hourly time steps for the different soil moisture products.**

| | | | | Daily | | | | Hourly | |
|---|---|---|---|---|---|---|---|---|---|
| | ASCAT | ESA-CCI | SMOS | SMOS-IC | ERA5 | In-situ | SMA 30mm | ERA5 | In-situ |
| | | | | | **RHERAYA** | | | | |
| Min | -0.15 | - | - | -0.04 | -0.73 | -1.88 | - | -0.01 | 0.15 |
| Mean | 0.48 | - | - | 0.58 | 0.45 | 0.25 | - | 0.64 | 0.54 |
| Median | 0.57 | - | - | 0.63 | 0.66 | 0.49 | - | 0.73 | 0.61 |
| Max | 0.85 | - | - | 0.84 | 0.82 | 0.83 | - | 0.81 | 0.71 |
| | | | | | **ISSYL** | | | | |
| Min | - | -56041 | -1938.07 | - | - | - | -96.08 | -114.6 | - |
| Mean | - | -14138.2 | -324.3 | - | - | - | -24.77 | -16.74 | - |
| Median | - | -254.85 | -1.8 | - | - | - | -2.46 | -0.85 | - |
| Max | - | -2.10 | -0.52 | - | - | - | -0.78 | 0.83 | - |


























**FIGURES**


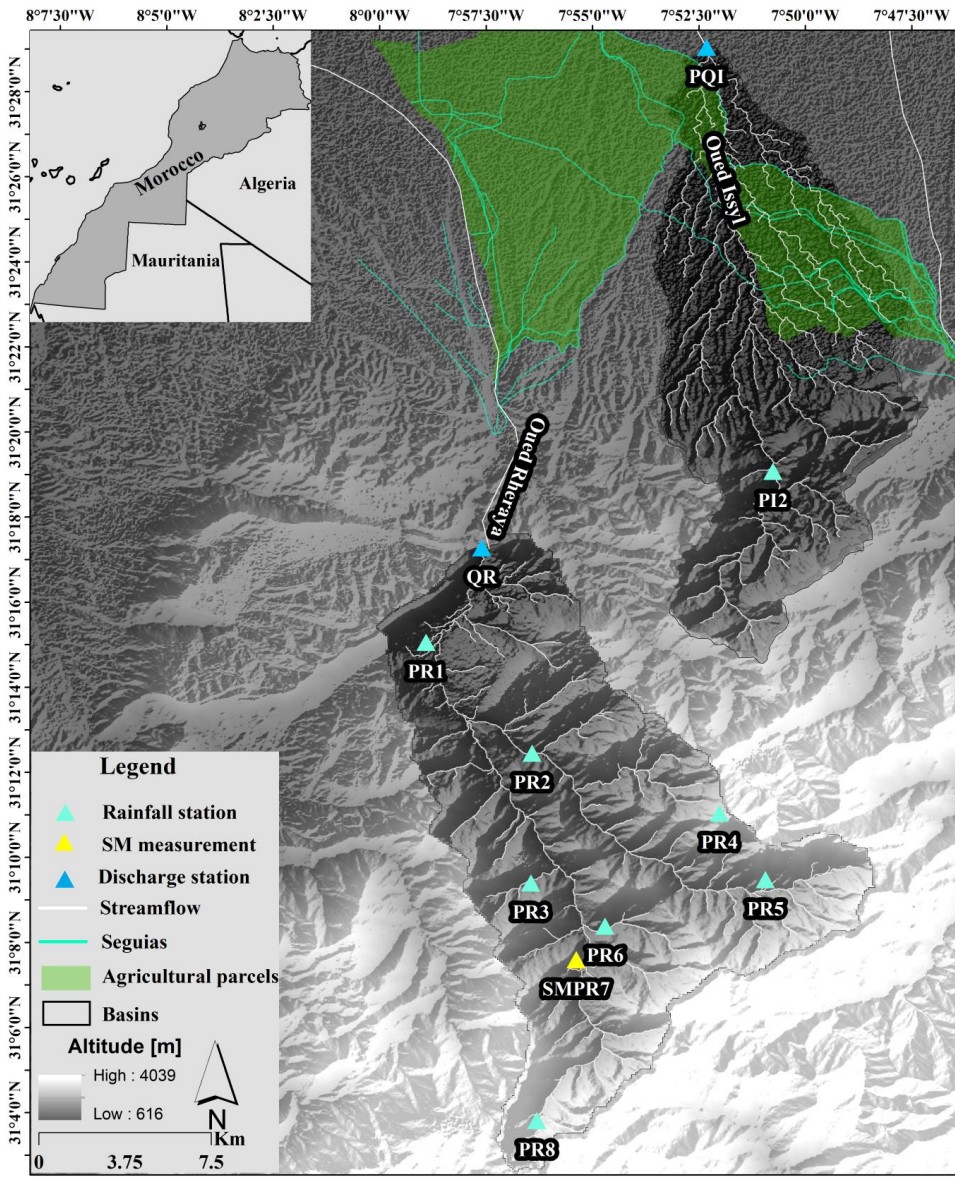


**Figure 1:  Location of Rheraya and Issyl basins, the seguias network, the agricultural parcels and the**
**hydro-meteorological network – PR: Rainfall station in Rheraya, SMPR: Soil moisture measurement+**
**Rainfall station in Rheraya, PQI: Rainfall and discharge station in Issyl, QR: Discharge station in**
**Rheraya.**


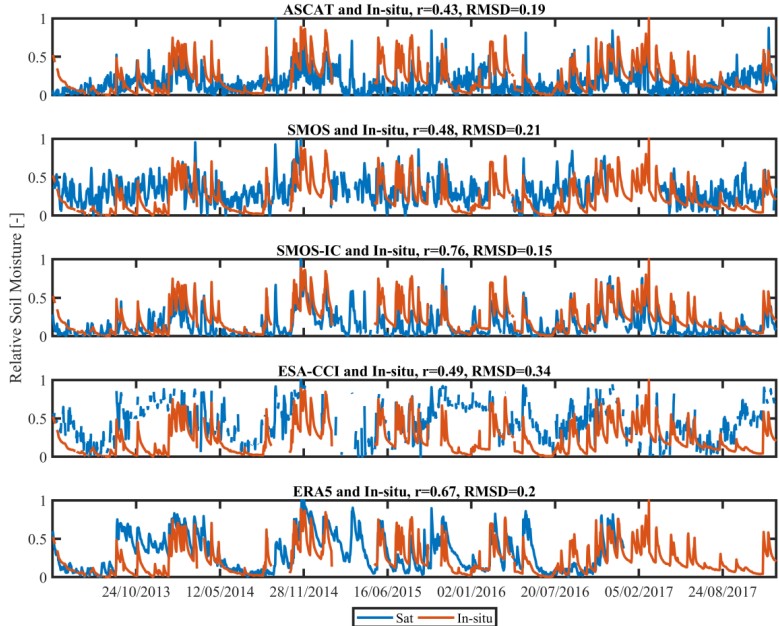


**Figure 2: Correlation between measurements of soil moisture (5cm depth) and different products of soil**
**moisture (Rheraya basin).**


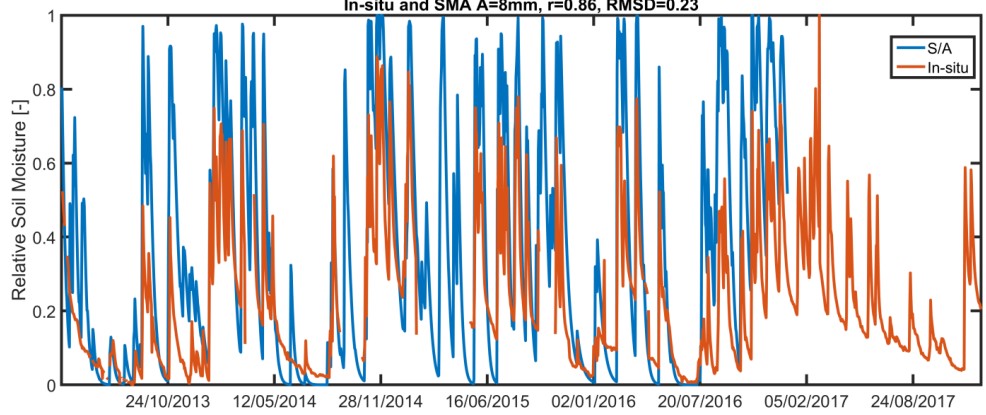

**Figure 3: Relationship between S/A and observed soil moisture data between 08/04/2013 and 31/12/2016**
**for different values of A (Rheraya basin).**

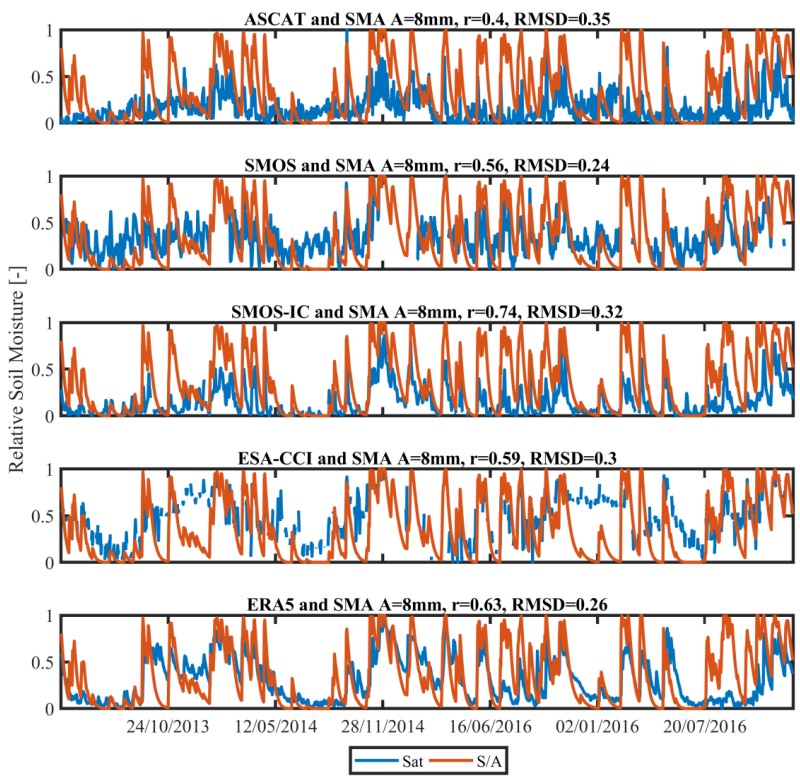


**Figure 4: Relationship between satellite products of soil moisture and ERA5 with and SMA outputs**

**between 08/04/2013 and 31/12/2016 over the Rheraya basin.**



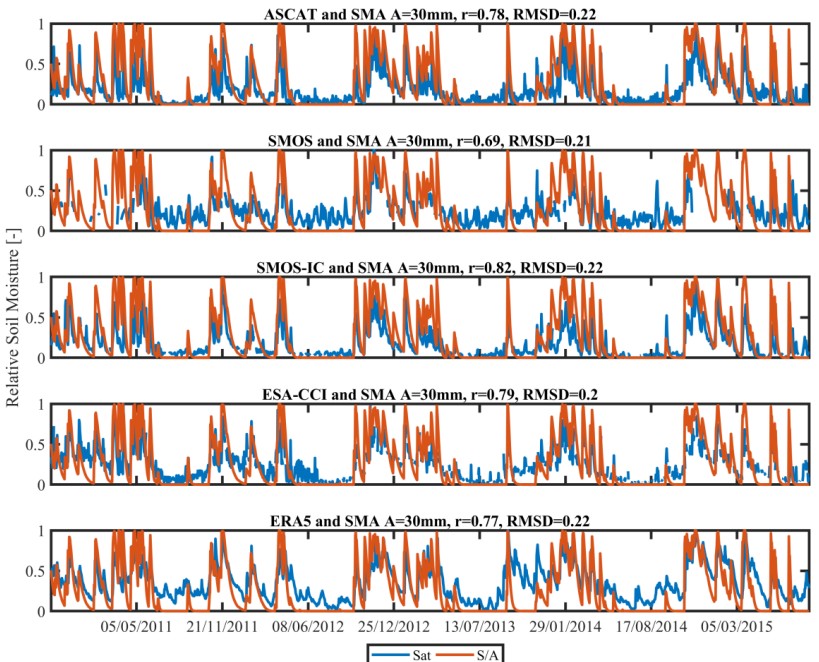

**Figure 5: Relationship between Satellite products of soil moisture and SMA outputs for A=30mm between 18/10/2010 and 20/08/2015 in the Issyl basin**

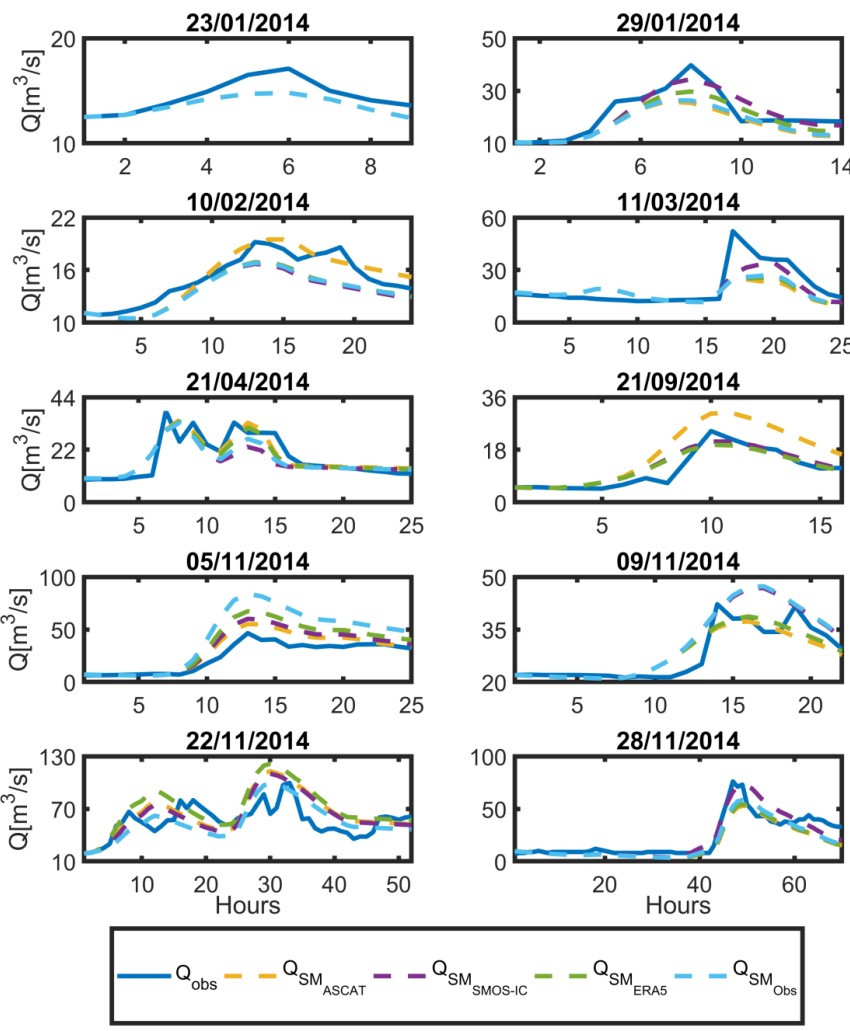

**Figure 6: Validation results of flood events simulated for the Rheraya using different soil moisture products with a daily time step. The observed hydrograph ($Q_{obs}$) is compared to the simulated hydrographs using ASCAT, SMOS-IC, ERA5 and in situ data.**



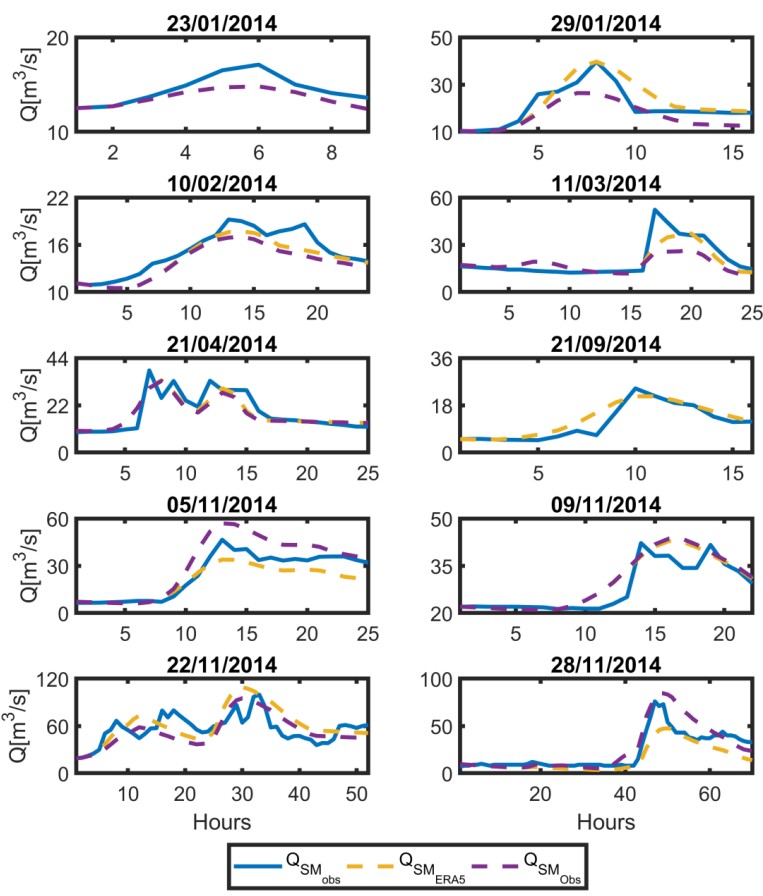


**Figure 7: Validation of the flood events simulated for the Rheraya using ERA5 and in situ soil moisture**

**with hourly time step.**

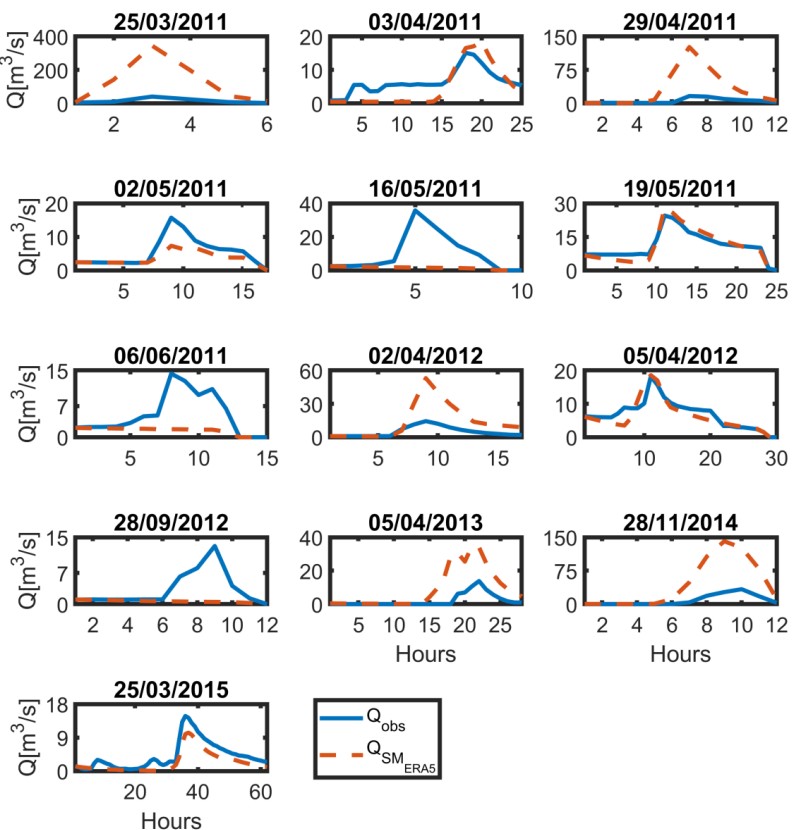


**Figure 8: Validation result of flood events for the Issyl using ERA5 with hourly time step**
