# Peer review of "Challenges in flood modeling over data scarce regions: how to exploit globally available soil moisture products to estimate antecedent soil wetness conditions in Morocco"

_Natural Hazards and Earth System Sciences, 2020_

## Author Comment (AC1) · 28 May 2020

We noticed that during the submission process two authors were not included by mistake : Vincent Simmoneaux and Simon Gascoin from CESBIO, Université de Toulouse, CNRS/CNES/IRD/INRA/UPS, 31400 Toulouse, France

They will be added during the review of the manuscript

---

## Referee Comment (RC1) · Anonymous Referee #1 · 9 Jun 2020

General comment

This paper is very interesting and very well written. It compares different soil moisture (SM) satellite products from an hydrological point of view. In particular, it tries to asses if these products could be useful in real time, as part of future flood warning system in Morocco. I think the paper could be published, after some minor changes.

Main remarks

1. The methods and data are globally well described, but sometimes a bit difficult

to follow due to the huge amount of information provided. I would see 2 tables: a table summarising the different available SM products, and another one concerning the punctual measurements in each catchments

2. I would also remove the Extended collocation analysis: it takes time to read and understand the method (3.2), while the results are presented in only few lines (4.4), confirming previous findings.

3. I agree with the authors concerning the usefulness of the SM satellite products. However, I don't think that they could totally replace a Soil Moisture Accounting (SMA) scheme, especially for real time flood forecasting. First, as mentioned by the authors, the latency of those satellite products could be an issue as well their coarse spatial resolution. My opinion is that an interesting perspective to mention is to assimilate SM satellite products into continuous models in order to correct the state of its production function. I am not sure that using an event-based hydrological model is a good option for (flash) flood forecasting. But I agree, this question is far beyond the scope of this paper.

Minor remarks

4. P02L68: no date in Western and Bloschl

5. P04L132: A table could summarise the punctual hydro-meteo data (type, number, starting date, ending date, time step, catchment, . . .)

6. P04L139: I think that with a such short observation period (1 year for Rheraya and 6 years for Issyl) you also have strong uncertainties on high flow, because of the rating curve. How the discharge were calculated? Are gauging during floods available?

7. P05L153: I don't know if the Oudin's formula has been tested before in Africa.

8. P06L187: "Hargreaves-Samani equation" => you said you were using the Oudin formula?

[Figure]

9. P05L189: A table could summarise these data (type, starting date, ending, time resolution, spatial resolution, latence,missing values. . .)

10. P07L252: replace "ranging between 1 and 1000" to "ranging from 1 to 1000mm"

11. P08L278: you must finish this paragraph by explaining the criteria r and RMSD (with in_situ and SMA as 'reference'

12. P08L279: remove this paragraph

13. P10L348: how is CN (and S) calibrated? On which criteria?

14. P11L355: same question for Sc and Tc

15. P11L368: I assume that Sc and Tc are also calculated using the leave-one-out procedure (as for S)

16. P11 equation (9) and (10): is N the number of time step, or the number of event? I think that here, this is the number of event, while previously (see remark 11), it is the number of time step

17. P12 equation (12): express le bias correctly (with sums)

18. P12L381: N => number of event or time step? + be coherent you also have 'n' in 10 and 11 (see rq 16)

19. P12L395: replace "from between 0,59 and 0,64" by "from 0,59 to 0,64"

20. P12L398-401: mention the % in the data table (see remark 9). But maybe, it is better to calculate the 'continues' r and RMSD, over a same time 'common' period, what ever the product you consider. Indeed, I think that the discrepancy in time period could have an impact on the scores.

21. P13L424: same remark

22. P14L457: delete 4.4 paragraph

23. P14L474: replace "table 3" by "table 4"

24. P16L537: this sentence should be in the introduction

25. P17L583: maybe replace "Javelle et al 2010", by "Javelle et al 2016: Setting up a French national flash flood warning system for ungauged catchments based on the AIGA method, DOI: 10.1051/e3sconf/20160718010

26. P17L587: additionally to the latency issue, there is also the spatial resolution issue. On this subject, see for instance this product: https://www.theia-land.fr/humidite-du-sol-a-thrs-cinq-series-mises-a-jour/

27. P17L579-590: see my remark 3. I think that in the future, we must investigate assimilating SM satellite data into continuous hydrological models.

28. P30 Table6: "-1938.07" should be on one line

29. Figure 2 to Figure 5: time scale with dash every 1rst January (to better see the seasonality)

30. Figure 7: replace in the legend "Q_SM_obs" by "Qobs"

---

## Referee Comment (RC2) · Anonymous Referee #2 · 7 Jul 2020

This paper focuses on the use of different globally available soil moisture products (satellite and reanalysis) to provide initial conditions for an event-based flood model. It is applied on two semi-arid catchments in Morocco and tries to evaluate the added value of these products for real-time flood forecasting in such environment. The manuscript is well written and organized, the methodology is clearly stated and the results convincingly lead to the authors conclusions. I think the paper is almost ready for publication.

I only have one main concern about the data used to force the model. The quality of

precipitation data used to force the model is not discussed, while it could highly impact the model performances. Also, it is not clear which precipitation data is used: it is from rain gages or radar or a combination of both? Given the results, it seems that radar observations are used to force the model. But then, how are used the rainfall stations presented in section 2.2? Are their observations compared to radar? On the other hand, evapotranspiration is also a crucial variable in semi-arid regions. Is the Oudin formula well suited for such environment?

Minor remarks:

L147. Could the authors remind the definition of the runoff coefficient?

L245. Does the SMA model account for any kind of spatial variability or is it just a simple lumped model?

L323. Please define sigma_theta and MSE.

L328. What does the # symbol mean?

L345. Is there any reason related to the model structure for the wide use of SCS-CN in semi-arid contexts? Also, I guess the SCS-CN model is a lumped hydrological model only simulating discharge at the outlet of the catchment. Is that correct?

L377. i and n in Eqs. (10-11) are not defined and could probably be simply removed, as in Eq. (9).

Figure 2. Please replace "Correlation" by "Comparison" in the figure caption. The figure does not show only correlations.

L410. The authors could show the differences between rainfall at site scale and catchment scale (the latter being used in the SMA model).

L421. Is there any possible explanation of the overestimation of soil moisture compared to in-situ measurements (e.g. lower rainfall at the in-situ site than over the entire catchment)?

L474. It is Table 4.

L475. "As shown on Figure 6, the SCS-CN model in calibration..." but "Validation" is written in the caption of Figure 6.

L478. Figure 7 is more likely discussed in section 4.6. Should it be Figure 8?

L486. Please explain (maybe at the end of section 3.3) why a highly negative correlation (close to -1) means that the simulation is good.

L524. Water uptakes during flood could explain the overestimation of the model compared to discharge observations (events 25/03/11, 29/04/11, 02/04/12, 05/04/13 and 25/11/14). But what could explain that the model completely missed the last three events (16/05/11, 06/06/11 and 28/09/12)?

---

## Author Response (AR1)

Dear Eric Martin,

We would like first to thank you for handling this manuscript. We adressed all the reviewers and your comments in the revised manuscript. Please find below the answers to your comments and then the response the the reviewer comments.

**Please submit a manuscript that take into account the reviwer's sugestions. In addition, I have the following remarks and suggestions :**
**1) As far as I understand, A represent the soil water content from the surface reservoir (near the surface) and is compared to satellite products that observed few cm of soil near the surface. I wel understand the 8 mm value for the Rheraya basin, but the 30 mm appear to me quite important and cover more than the "few cm" observed. Could you comment on that ?**

We added more detail line 201 of the revised manuscript: "… parameter A, which represents the maximum reservoir capacity of the soil".

The SMA model considered is the same as in Javelle et al., 2010, corresponding to the production store of the GR4J model (Perrin et al, 2003). It represents, averaged over the catchment, the rainfall depth in mm that the basin could store before initiating surface runoff. It cannot be directly related to soil depth. The optimal values found for this A parameter are fully consistent with previous studies in Morocco (Tramblay et al., 2012); with values much lower than in other regions due to lower soil moisture storage capacity in these semi-arid basins of Morocco. For comparison, the median value for this parameter in Perrin et al., 2003 for many catchment in France and other temperate zones is 350 mm.

**2) Parapgraph 4.3 (comparison by seasons).**
**Line 453 : the term "by moving from winter to autumn" is not precise. There is in both basin a minimum in summer and the autumn is not the maximum for the Reraya. Please change the wording here and also in the conclusion.**

We agree that this sentence is not clear. Here we analyze the results of the mean correlation per season averaged for all products, to have a more robust signal. We replaced this sentence line 491 by:

"The highest mean correlations (i.e. averaged for all the different products) are found during fall in the Rheraya basin, with r=0.61 with in situ data and r=0.58 with SMA soil moisture. It should be noted that correlations with SMA outputs in summer are similar with r=0.59. For the Issyl basin, the correlations are also higher in the fall with a mean r=0.87 for the SMA model."

We also change the sentence line 591 in the conclusion: "The seasonal analysis showed for both basins that the highest correlations are found in autumn, which encourages the use of these remote sensing products for flood forecasting because the majority of events occur in autumn and early winter in these regions"

**3) Table and figure legends**

**Please check all Table and Figure Legend. Some are very short (e.g Table 1) or wrong (Figure 2 does not show correlation, but evolution). The legends must be as far as possible comprehensive, so that the reader can understand it without refereing immediately to the text.**

We changed the caption of Table 1 to: "Stations with observed precipitation and river discharge"

Also for Table 5: Results of the correlation analysis between the different soil moisture data, in-situ measurements and SMA model outputs (significant correlations are represented in bold)

We changed the caption of figure 2 to: "Comparison between measurements of soil moisture (5cm depth) and different products of soil moisture (Rheraya basin)."

Minor spelling errors have also been modified in other figure captions.

**4) Tables : be careful with numbers : some correlation are writen with 2 digits, some others with one. Please replace 0.2 or 0.3 by 0.20 or 0.30.**

Changed

**5) Figure 1 : The grey color in the figure does not correspond to elevation only as stated in the legend. There is also an arientation component. Please check.**

We modified figure 1 to show only the elevation.

**Referee #1**

**General comments:**

**This paper is very interesting and very well written. It compares different soil moisture (SM) satellite products from a hydrological point of view. In particular, it tries to asses if these products could be useful in real time, as part of future flood warning system in Morocco. I think the paper could be published, after some minor changes.**

> *First of all, we would like to thank the reviewer for his time to critically review our manuscript and to provide additional suggestions to improve the paper. In this document, we respond to the received comments point by point and also we show the changes suggested in the paper according to the line numbers in the revised document. Our responses are in italics.*

**Main remarks:**

1. **The methods and data are globally well described, but sometimes a bit difficult to follow due to the huge amount of information provided. I would see 2 tables: a table summarizing the different available SM products, and another one concerning the punctual measurements in each catchments**

   > *Thanks for this comment. We will follow your suggestion and we added a table that summarizes the different rain gauges used in the paper and also the hydrometric stations with different information (Time step, monitoring period, altitude…).*

   > *Concerning the second table, that presents the different available SM products, we added a table so the reader can have a clear view of the soil moisture products considered.*

2. **I would also remove the Extended collocation analysis: it takes time to read and understand the method (3.2), while the results are presented in only few lines (4.4), confirming previous findings**

   > *We added the Extended Collocation analysis to verify if the two types of comparisons give the same results. But we will follow your suggestion and this part has been removed in the revised manuscript. We added this evaluation as supplementary material.*

3. **I agree with the authors concerning the usefulness of the SM satellite products. However, I don't think that they could totally replace a Soil Moisture Accounting (SMA) scheme, especially for real time flood forecasting. First, as mentioned by the authors, the latency of those satellite products could be an issue as well their coarse spatial resolution. My opinion is that an interesting perspective to mention is to assimilate SM satellite products into continuous models in order to correct the state of its production function. I am not sure that**

**using an event-based hydrological model is a good option for (flash) flood forecasting. But I agree, this question is far beyond the scope of this paper**

> *We agree with the reviewer, we cannot replace easily SMA model which is based on observed data with satellite products especially for real time flood forecasting. However the accuracy of the SMA model depends on the quality of rainfall data. As the reviewer mentioned, we can mention in the perspective the contribution of SM product assimilation in order to correct the data [Line: 615 to 620). Another option would be to calibrate the SMA product with satellite data (as in Tramblay et al. 2012).*

> *About the use of a continuous model, its application can be hampered by the lack of long-term good quality data. In particular rainfall but also runoff, since stage/discharge relationship may change over time due the changes in the river channel in this type of mountainous basins. This is why we implemented an event-based approach.*

**Minor remarks:**

4. **P02L68: no date in Western and Bloschl**

   > *Done.*
* * *
5. **P04L132: A table could summarize the punctual hydro-meteo data (type, number, starting date, ending date, time step, catchment, . . .)**

   > *Done.*
* * *
6. **P04L139: I think that with a such short observation period (1 year for Rheraya and 6 years for Issyl) you also have strong uncertainties on high flow, because of the rating curve. How the discharge were calculated? Are gauging during floods available?**

   > *The discharge was calculated using the radar sensor installed in each basin's outlet with a time step of 10min. The radar observes the height of the flow and then the discharge is calculated from the rating curve by the Hydraulic Agency of Tensift. The rating curves have been elaborated on much longer records than 1 and 6 years, but due to data quality issues, some events did not have paired rainfall and runoff, or some floods were obviously erroneous, we performed a conservative selection of flood events to ensure the quality of the data.*

   > *We don't know in detail the gauging strategy of the Hydraulic agency, but due to the presence of a bridge at the location of the gauge, it is possible to measure river speed event at relatively high discharge rates.*
* * *
7. **P05L153: I don't know if the Oudin's formula has been tested before in Africa.**

*The Oudin formula was previously applied tested in Morocco (Tramblay et al., 2013, Marchane et al., 2017) and in Tunisia (Dakhlaoui et al., 2020). We added this sentence from line 163 to 165.*
* * *
8. **P06L187: "Hargreaves-Samani equation" => you said you were using the Oudin formula?**

   *Yes, we made a mistake there. We corrected it.*
* * *
9. **P05L189: A table could summarise these data (type, starting date, ending, time resolution, spatial resolution, latence,missing values. . .)**

   *We added this new table as table 3*
* * *
10. **P07L252: replace "ranging between 1 and 1000" to "ranging from 1 to 1000mm"**

    *Done.*
* * *
11. **P08L278: you must finish this paragraph by explaining the criteria r and RMSD (with in_situ and SMA as 'reference'**

    *Done, we completed the paragraph that explains the r and RMSD criteria. From line 299 to 300 "With $SM_{In-situ}$ is the in-situ measurements of soil moisture or SMA model which are considered as reference, $SM_{sat}$ is the soil moisture from satellite or reanalysis and N is the number of values."*
* * *
12. **P08L279: remove this paragraph**

    *Done.*
* * *
13. **P10L348: how is CN (and S) calibrated? On which criteria?**

    *The parameter CN is first calibrated both automatically in the HEC-HMS software and manually; to obtain the correct shape of the hydrograph, and then we calibrate the other parameters that condition the transfer (Sc and Tc), that mean that the calibration is made separately between the Production and Transfer functions. The main criterion is Nash-Sutcliffe efficiency coefficient.*

    *These explanations have been added from line 374-384.*
* * *
14. **P11L355: same question for Sc and Tc**

   *The answer is mentioned in the previous point*
* * *
15. **P11L368: I assume that Sc and Tc are also calculated using the leave-one-out procedure (as for S)**

   *Yes. In the leave-one-out procedure, the model is recalibrated with the N-1 events, then the mean Sc and Tc values are used in validation. However, there is a difference between the calculation of the S parameter and the other two. The difference is that the calculation of S is based on a linear equation that links it with SM data.*
* * *
16. **P11 equation (9) and (10): is N the number of time step, or the number of event? I think that here, this is the number of event, while previously (see remark 11), it is the number of time step**

   *The displacement of those equations as you suggested in the point 11 resolve this problem.*
* * *
17. **P12 equation (12): express le bias correctly (with sums)**

   Done.
* * *
18. **P12L381: N => number of event or time step? + be coherent you also have 'n' in 10 and 11 (see rq 16)**

   *Done.*
* * *
19. **P12L395: replace "from between 0,59 and 0,64" by "from 0,59 to 0,64"**

   *Done*.
* * *
20. **P12L398-401: mention the % in the data table (see remark 9). But maybe, it is better to calculate the 'continues' r and RMSD, over a same time 'common' period, whatever the product you consider. Indeed, I think that the discrepancy in time period could have an impact on the scores.**

   *We added the % in the table. We calculate the r and RMSD over the same period. But in the figure presentation we showed the whole period.*

**21. P13L424: same remark**

*Done.*

**22. P14L457: delete 4.4 paragraph**

*Done.*

**23. P14L474: replace "table 3" by "table 4"**

*Done.*

**24. P16L537: this sentence should be in the introduction**

Done.

**25. P17L583: maybe replace "Javelle et al 2010", by "Javelle et al 2016: Setting up a French national flash flood warning system for ungauged catchments based on the AIGA method, DOI: 10.1051/e3sconf/20160718010**

*We replaced it. Thank you.*

**26. P17L587: additionally to the latency issue, there is also the spatial resolution issue. On this subject, see for instance this product: https://www.theia-land.fr/humidite-du-sola-thrs-cinq-series-mises-a-jour/**

*We totally agree with the reviewer, the spatial resolution is an issue in this type of products. We added page 18, line 618 to 620: "With the issue of the latency to obtain some products, it should be noted also that the mismatch of spatial resolution between large scale remote sensing products and very local small scale applications could be an additional issue".*

**27. P17L579-590: see my remark 3. I think that in the future, we must investigate assimilating SM satellite data into continuous hydrological models.**

*Yes we added this point.*

**28. P30 Table6: "-1938.07" should be on one line**

    *Done.*
* * *
**29. Figure 2 to Figure 5: time scale with dash every 1rst January (to better see the seasonality)**

    *We changed the time scale to show every 1st January and 1st June of each year.*
* * *
**30. Figure 7: replace in the legend "Q_SM_obs" by "Qobs"**

    *Done.*

**Referee # 2**

This paper focuses on the use of different globally available soil moisture products (satellite and reanalysis) to provide initial conditions for an event-based flood model. It is applied on two semi-arid catchments in Morocco and tries to evaluate the added value of these products for real-time flood forecasting in such environment. The manuscript is well written and organized, the methodology is clearly stated and the results convincingly lead to the authors conclusions. I think the paper is almost ready for publication.

> *We would like to thank the reviewer for reading our work and for providing important suggestions in order to improve the paper.*
* * *
I only have one main concern about the data used to force the model. The quality precipitation data used to force the model is not discussed, while it could highly impact the model performances

> *We agree. The observed precipitation quality was not well discussed, we added in the revised manuscript a description from line 144 to 149: "The precipitation data is missing in some events, especially at high altitude gauges during snowfall events. The percentage of missing value ranges from 2.4% at PR5 to 10.85% at PR7. In other hand, the highest percentage of 19.7% is found at PR1 where the gauge underwent technical problems. Overall, the total percentage of missing value (7.8%) is very low, hence no filling method is used ». A new table also list the data available.*
* * *
Also, it is not clear which precipitation data is used: it is from rain gages or radar or a combination of both? Given the results, it seems that radar observations are used to force the model. But then, how are used the rainfall stations presented in section 2.2? Are their observations compared to radar?

> *We used in this study rainfall gauges not radar. The radar that we indicate in the line 138 in the section 2.2 is related to the hydrometric data that is measured using a radar sensor in each basin's outlet. No meteorological radar is available in this region.*
* * *
On the other hand, evapotranspiration is also a crucial variable in semi-arid regions. Is the Oudin formula well suited for such environment?

> *We based our choice on the study of Marchane et al., 2017 on the same basin who compares different equations of evapotranspiration and it is concluded that Oudin estimates are very comparable to other formulas (Hargreaves-Samani and FAO-Penman).*

**Minor remarks:**

1. **L147. Could the authors remind the definition of the runoff coefficient?**

   *We added the definition from line 155 : 157*
* * *
2. **L245. Does the SMA model account for any kind of spatial variability or is it just a simple lumped model?**

   *The SMA model is a lumped model but the daily precipitation had been interpolated over the basin to obtain the mean areal precipitation*
* * *
3. **L323. Please define sigma_theta and MSE**

   *We followed the suggestion of the 1st reviewer and we deleted the entire section*
* * *
4. **L328. What does the # symbol mean?**

   *We deleted this section as the 1st reviewer suggested.*
* * *
5. **L345. Is there any reason related to the model structure for the wide use of SCS-CN in semi-arid contexts? Also, I guess the SCS-CN model is a lumped hydrological model only simulating discharge at the outlet of the catchment. Is that correct?**

   *The widely use of SCS-CN model is related to its simplicity and low number of parameters and also because it requires only rainfall and discharge to simulate runoff at the outlet of the catchment. It has been widely applied in the Mediterranean region. But indeed it is not a model specifically tailored for semi-arid areas.*
* * *
6. **L377. i and n in Eqs. (10-11) are not defined and could probably be simply removed, as in Eq. (9).**

   *We deleted them.*
* * *
7. **Figure 2. Please replace "Correlation" by "Comparison" in the figure caption. The figure does not show only correlations**

   *Done.*
* * *
8. **L410. The authors could show the differences between rainfall at site scale and catchment scale (the latter being used in the SMA model).**

> *We do not fully understand this comment. We used data from rain gauges to interpolate rainfall over the whole catchment and compute daily areal rainfall. The use of basin-averaged rainfall is also a good way to smooth out the uncertainties related to individual rain gauges.*

9. **L421. Is there any possible explanation of the overestimation of soil moisture compared to in-situ measurements (e.g. lower rainfall at the in-situ site than over the entire catchment)?**

> *Yes, the location of the soil moisture sensors is probably not representative of the soil type and precipitation amounts of the whole catchment. Indeed, soil moisture probes are located at about 2000 m.a.s.l. and with steep slopes, whereas downstream parts of the basin may have deeper soils able to store more soil moisture.*

10. **L474. It is Table 4**

> *Yes it is Table 4, Thank you.*

11. **L475. "As shown on Figure 6, the SCS-CN model in calibration..." but "Validation" is written in the caption of Figure 6**

> *We deleted ''Figure 6'', thank you.*

12. **L478. Figure 7 is more likely discussed in section 4.6. Should it be Figure 8?**

> *Thank you, we replaced the discussion of the figures 6, 7 and 8 into the section 4.6.*

13. **L486. Please explain (maybe at the end of section 3.3) why a highly negative correlation (close to -1) means that the simulation is good**

> *We added this explanation in the text from line 390 to 392: "The relationship is good when the correlation is near to r= -1. The negative correlation is related to the fact that, the storage capacity (S) is larger when the soil is dry (soil moisture is near to 0) and vice versa".*

14. **L524. Water uptakes during flood could explain the overestimation of the model compared to discharge observations (events 25/03/11, 29/04/11, 02/04/12, 05/04/13 and 25/11/14). But**

**what could explain that the model completely missed the last three events (16/05/11, 06/06/11 and 28/09/12)?**

*Yes the events 16/05/2011and 06/06/2011 showed an important spatial variation of precipitation with no precipitation observed 
[revised manuscript text omitted]

---

## Author Response (AR2)

Dear Eric Martin,

We addressed your comments, please see the revised manuscript and our comments below.

**Line 495 : suppress a ","**

Done

**Lines 501- 508 : Please reword and simplify these sentences (it is quite wordy at this stage)**

We reworded this section.

**Lines 536-537 : I don't understand the sentence beginning by "However…" . There are assimilation techniques (most of them) that can use data that are not continuous. On the other way, of course, if the series are too discontinuous, with too much missing data the assimilation won't be efficient, but that is trivial. Please check this sentence.**

Indeed, we agree that sentence was quite trivial. We removed this sentence.

**Line 868 : add the period considered in the legend**

We added "between 2013 and 2016"

**Line 871 : modify the legend : "Seasonal correlation between the different soil moisture data, in situ measurements and the SMA model (significant correlations are represented in bold)"**

Changed

**Legend of Fig. 4 and Fig. 5 : please homogenise the two legend (Add "A" in Fig 4 or delete in Fig 5.). Add into bracket "Sat" and "S/A" in the caption. E.g. Relationship between the different product of soil moisture (Sat) and SMA outputs (S/A)….**

We homogenized the captions:
Figure 4: Relationship between the different products of soil moisture and SMA outputs between 08/04/2013 and 31/12/2016 over the Rheraya basin.
Figure 5: Relationship between the different products of soil moisture and SMA outputs between 18/10/2010 and 20/08/2015 in the Issyl basin

**Figure 6 : Precise the caption : Validation results of flood events simulated for the Rheraya using different soil moisture products with a daily time step : the observed hydrographs ($Qobs$) are compared to the simulated hydrographs using ASCAT (Qascat), SMOS-IC (QSMOS-IC) …. For the selected flood events described in Table 2**

**Figure 7 : see comment on Figure 6**

**Figure 8 : homogenise the caption with Fig. 5 and 6.**

We modified as follow:

[revised manuscript text omitted]